# *Leishmania naiffi* and *lainsoni* in French Guiana: Clinical features and phylogenetic variability

**Océane Ducharme**[1], **Stéphane Simon**[2], **Marine Ginouves**[2], **Ghislaine Prévot**[2], **Pierre Couppie**[1,2,3], **Magalie Demar**[2,3,4], **Romain Blaizot**[1,2,3]*

**1** Service de Dermatologie, Hôpital Andrée Rosemon, Cayenne, French Guiana, **2** Equipe EA3593, Ecosystèmes Amazoniens et Pathologie Tropicale, Université de la Guyane, Cayenne, French Guiana, **3** Centre National de Référence des Leishmanioses, laboratoire associé, Hôpital Andrée Rosemon, Cayenne, French Guiana, **4** Laboratoire Hospitalo-Universitaire de Parasitologie-Mycologie, Hôpital Andrée Rosemon, Cayenne, French Guiana

* blaizot.romain@gmail.com

**Data Availability Statement:** All relevant data are within the manuscript and its Supporting Information files.

## Abstract

In French Guiana, five species are associated with Cutaneous Leishmaniasis (CL). Though infections with *Leishmania guyanensis*, *L. (V.) braziliensis* and *L. (L.) amazonensis* have been extensively described, there are few available clinical and genetic data on *L. (V.) lainsoni* and *L. (V.) naiffi*. We determined the clinical and epidemiological features of all cases of CL due to *L. (V.) naiffi* and *L. (V.) lainsoni* diagnosed in French Guiana between 2003 and 2019. Phylogenetic analysis was performed by sequencing a portion of *HSP70* and *cyt b* genes. Five cases of *L. naiffi* and 25 cases of *L. lainsoni* were reported. Patients infected by *L. (V.) lainsoni* were usually infected on gold camps, mostly along the Maroni river (60%), while *L. naiffi* was observed in French patients infected on the coast (100%). A high number of pediatric cases (n = 5; 20%) was observed for *L. (V.) lainsoni*. A mild clinical course was observed for all cases of *L. (V.) naiffi*. *HSP70* and *cyt b* partial nucleotide sequence analysis revealed different geographical clusters within *L. (V.) naiffi* and *L. (V.) lainsoni* but no association were found between phylogenetic and clinical features. Our data suggest distinct socio-epidemiological features for these two *Leishmania* species. Patients seem to get infected with *L. (V.) naiffi* during leisure activities in anthropized coastal areas, while *L. (V.) lainsoni* shares common features with *L. (V.) guyanensis* and *braziliensis* and seems to be acquired during professional activities in primary forest regions. Phylogenetic analysis has provided information on the intraspecific genetic variability of *L. (V.) naiffi* and *L. (V.) lainsoni* and how these genotypes are distributed at the geographic level.

## Author summary

Cutaneous leishmaniasis is a parasitic disease affecting at least 12 million people in 96 countries. In French Guiana, five species of Leishmania are involved in human disease: *Leishmania (V.) guyanensis*, *Leishmania (V.) braziliensis* and *Leishmania (L.) amazonensis* are common and have been extensively studied. *Leishmania (V.) lainsoni* and *Leishmania*

**Funding:** The author(s) received no specific funding for this work.

**Competing interests:** The authors have declared that no competing interests exist.

(V.) naiffi are less frequent and very few data are available on the patients infected by these species. In this study, we identified five cases of human patients infected by *L. (V.) naiffi* and 25 cases of *L. (V.) lainsoni*. Patients infected by *L. (V.) lainsoni* were usually god miners infected in the rainforest, while *L. naiffi* was observed in patients infected on the anthropized coast of French Guiana. *L. naiffi* was associated with mild lesions. Pentamidine was an efficient treatment for most cases of both species.

## Introduction

Leishmaniasis is considered as a "neglected tropical disease" by the World Health Organization [1], affecting at least 12 million people in 98 countries [2]. It is caused by parasites of the *Leishmania* genus. *Leishmania* flagellates are grouped into two sub-genera, *Leishmania* and *Viannia*, according to their development in the insect digestive tract.

In French Guiana, five species of Cutaneous Leishmaniasis (CL): *Leishmania (Viannia) guyanensis*, *Leishmania (Viannia) braziliensis*, *Leishmania (Leishmania) amazonensis*, *Leishmania (Viannia) lainsoni* and *Leishmania (Viannia) naiffi*, are associated with CL and mucocutaneous leishmaniasis (MCL) [3–5]. Clinical presentations depend on several factors including infecting species and host immunological response. *L. (V.) guyanensis* and *L. (V.) braziliensis*, the predominant species in French Guiana, have been well documented in the literature [3,5–7]. *L. (V.) braziliensis* is a virulent species potentially responsible for mucous involvement, although *L. (V.) guyanensis* has also been incriminated in diffuse clinical presentation and mucosal involvement, to a lesser extent [5,8].

However, *L. (V.) lainsoni* and *L. (V.) naiffi* have been rarely described and the clinical features of infections patients were not thoroughly studied. CL due to *L. (V.) naiffi* was described for the first time in Brazil, in Para state, in 1989 [9,10] and has been isolated from skin and viscera of nine-banded armadillo (*Dasypus novemcinctus*) [9,11]. Since then, only few cases or small series have been reported [12–15]. These reports described a benign clinical course with spontaneous healing or a good outcome with pentamidine. However, recent reports suggest that *L. (V.) naiffi* could be more severe and resistant to first-line treatment, with a poor response to meglumine antimoniate or pentamidine [16,17].

CL due to *L. (V.) lainsoni* was first described in 1987 [18] and seems to be widely distributed in South America with cases reported in Brazil, Peru, Bolivia, Surinam, French Guiana and recently in Colombia and Equator [5,19–23] Several studies reported an atypical phylogenetic profile with a distinct lineage from other species of the *Viannia* subgenus [24–26]. *L. (V.) lainsoni* is also characterized by the ease of cultivation in simple axenic culture media compared to other *Leishmania* from *Viannia* sub-genera [26]. Similarly to *L. (V.) naiffi*, it appears that *L. (V.) lainsoni* causes small ulcers or nodules [18,26]. However, these data are scarce and deserve further investigations.

This present study aimed to evaluate the clinical features and distribution of *L. (V.) naiffi* and *L. (V.) lainsoni* in French Guiana. Our second objective was to examine intra-species genetic variability by phylogenetic analysis, to look for associations between genetic polymorphism and clinical or epidemiological features.

## Materials and methods

### Ethics statement

According to the French public Health and Bioethical Law by the time of the study period, no ethics committee approval was needed for this observational study on routinely collected

biological material (Article L1121-1 and Article R1121-3). This study used only health care data that are routinely used for clinical purposes and is part of the usual research works of the National Reference Center for Cutaneous Leishmaniasis. The identification of *Leishmania* species is part of the surveillance and alert missions of the National Reference Center (CNR) of *Leishmania*. All patients were informed (leaflets, posters in several local languages) that data may be used for research and scientific publications, and that they had a right to refuse. All human data were anonymized.

## Inclusion criteria

We conducted an observational, retrospective study across French Guiana. All patients with proven diagnosis of *L. (V.) lainsoni* or *L. (V.) naiffi* infections between 01/01/2003 and 01/05/2019 were included. Data were extracted from the database belonging to the Parasitology Laboratory of the Andrée Rosemon Hospital, which is the referral center for diagnosis of CL in French Guiana. The study was conducted in the Cayenne General Hospital and in the health centers for remote areas (**Fig 1, S1 checklist**).

A new case was defined by the absence of documented history of leishmaniasis in the twelve months preceding the date of consultation. Positive diagnosis was defined by positive smear or culture, or positive DNA amplification during PCR-RFLP on skin biopsies or Hsp70 PCR on swab samples. Biopsies were cultivated in RPMI medium during up to 21 days at 28˚C. Species identification was then made with species-specific bands on PCR-RFLP [5,27] and/or Hsp70 sequencing [28] and/or with MALDI-TOF on positive cultures [29].

## Study variables

Study variables were collected from medical records of the Dermatology and Parasitology departments of the Cayenne hospital as follows: age, sex, birth and living places, site of possible contamination, profession and clinical factors: immunodeficiency, clinical presentation,

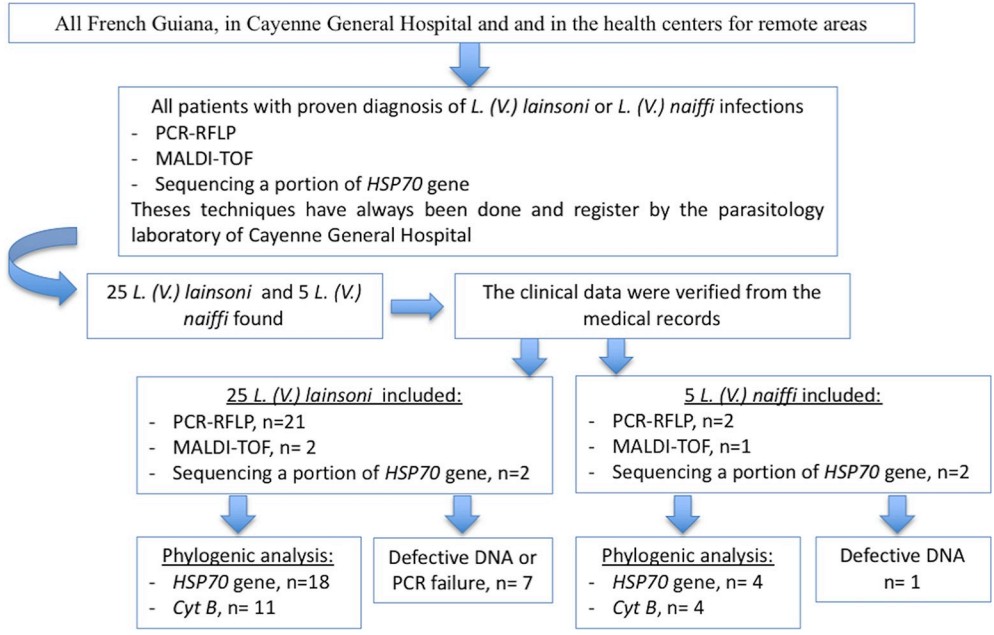

**Fig 1. Flow-chart of the study.**

number and diameter of lesions, time of evolution before diagnosis, lymphadenopathy and treatment. Places of contamination were classified as in previous studies, by dividing French Guiana in four main regions [5]: Coastal region, Maroni region, Center region and Oyapock region.

### *Leishmania* species identification

Due to the presence of several *Leishmania* species in French Guiana and the clinical implications of species identification, several techniques have been developed during the last decade to improve the speed and precision of parasitic diagnosis. Since 2005, the development of a PCR-RFLP technique targeting a 615-bp region of the RNA polymerase II gene has allowed a precise species identification of most cases [5,27]. Since 2016, another technique based on MALDI-TOF has been developed and allows species identification on positive skin biopsy cultures, even in case of negative PCR-RFLP [28]. Since 2018, PCR-RFLP has been replaced by sequencing of heat-shock protein 70 (*HSP70*) after PCR on swab samples [29]. Sampling with swabs replaced skin biopsies after 2018, as a superior sensitivity was established in several studies throughout South America [30–32].

A retrospective research was performed to retrieve all samples of *L. (V.) naiffi* and *L. (V.) lainsoni* identified since 2007 with any of these three techniques (**Fig 1**).

### DNA extraction and PCR amplification for *Leishmania* identification

DNA extraction was performed using QIAamp DNA Mini Kit (QIAGEN, Hilden, Germany), according to the manufacturer's instructions. Primers Hsp70sen (5'-GACGGTGCCTGCCT ACTTCAA-3') and Hsp70ant (5' CCGCCCATGCTCTGGTACATC 3') [28] were used to amplify *HSP70* fragment. PCR conditions were 95˚C for 5 min. following by 35 cycles of 95˚C for 1min, 60˚C for 1min and 72˚C for 1.5min, with a final extension at 72˚C for 5min. For the *cyt b* gene, LCBF1 (5'-GGTGTAGGTTTTAGTTTAGG3') and LCBR2 (5'-CTACAATAAA-CAAATCATAATATACAATT-3') [33] were used with the following PCR conditions: 95˚C for 5 min. and 35 cycles of 95˚C for 1min, 58˚C for 1min and 72˚C for 1.5min, with a final extension at 72˚C for 5min. PCR products were checked using the 2100 Bioanalyzer (Agilent Technologies, Santa Clara, USA) and were sent to Eurofins (Ivry sur seine, France) for sequencing. Obtained gene sequences were subjected to BLASTn on GenBank (NCBI web site) to search similarity with the *Leishmania* sequences.

### Sequences alignment and *HSP70* and *cyt b* partial nucleotide sequence analysis

The sequences were aligned by using CLUSTALX 2.1 sofware. Using MEGA software 7.0 (Penn State University, PA, USA), a phylogenetic tree was then built for each gene. Distances from nucleotide sequences were estimated with the Kimura-2 parameter model [34], and trees were built with the maximum likelihood (ML) method and bootstrap resampling was used across 1000 replicates.

The database for phylogenetic analyses consisted of *HSP70* gene sequences from *L. (V.) lainsoni* (GenBank accession number: FN395047; FN395049; LN907839; GU071179; FN395050), *L. (V.) naiffi* (FR872767; GU071183; FN395056; KX573968), *L. (V.) guyanensis* (FN395052), *L. (V.) braziliensis* (FN395043) *L. (L.) donovani* (KX061893), *L. (L.) infantum* (JX021433), *L. (L.) tropica* (KX061899), *L. (V.) peruviana* (EU599089), *L. (L.) amazonensis* (EU599090), *L. (L.) major* (HF586346), *T. cruzi* (KC959990).

The database for phylogenetic analyses consisted of *cyt b* gene sequences from *L. (V.) lainsoni* (GenBank accession number: LC153271), *L. (V.) naiffi* (LC153257), *L. (V.) guyanensis*

(AB095969), *L. (V.) braziliensis* (AB095967), *L. (V.) panamensis* (AB095968), *L. (L.) amazonensis* (EF579902), *T. vivax* (KM386446).

Lastly, sequence genetic diversity and haplotype diversity were calculated for *cyt b* and *HSP70* genes using DNAsp v.5.0 [23].

## Statistical analysis

The relationship between clinical-demographic variables and the infecting *Leishmania* species was analyzed using Fisher's exact test or $\chi^2$ test, as appropriate.

Statistical analyses were performed using GraphPad Prism software (v 6.01, San Diego, USA) with a significance bilateral threshold of 0.05.

## Results

### *Leishmania* species identification:

Between 01/01/2003 and 01/05/2019, five cases of *L. (V.) naiffi* and 25 cases of *L. (V.) lainsoni* were diagnosed in French Guiana. Concerning *L. (V.) naiffi*, all cultures were positive and the initial species identification was made by PCR-RFLP in two patients, MALDI-TOF in one of them and by *HSP70* gene sequencing in two others (**Fig 1**).

Concerning *L. (V.) lainsoni*, 14 (56%) cultures were positive, 7 were negative and this information was missing for four patients. Species identification was performed by PCR-RFLP for 21 patients, by MALDI-TOF for two and by sequencing a portion of *HSP70* gene for two patients (**Fig 1**).

### *L. (V.) naiffi* and *L. (V.) lainsoni* had distinct clinico-epidemiological features (**Table 1**)

Between 2003 and 2019, thirty patients were included: five with a proven diagnosis of *L. (V.) naiffi* and twenty-five of *L. (V.) lainsoni*. The median age of the 30 patients at diagnosis was 33,5 years [range 0 to 63] and the sex ratio was 1,5 (18 men/12 women). Most patients were born in Brazil (n = 17, 57%), followed by patients born in French Guiana (n = 5, 17%) and French mainland (n = 5, 17%). Eight of the 30 patients (27%) patients had an outdoor occupational profession, 20% were gold miners, 3% foresters and 3% military. For nine patients (30%), professional data were missing. Concerning *L. (V.) naiffi*, none of the five-recorded patients worked as a gold miner. Only one of them belonged to the military and was likely to work in primary forest. In total, the Maroni region was the most frequent site of probable contamination (n = 15, 50%), followed by the coastal region (n = 9, 30%), then the center region (n = 4, 13%) and the Oyapock region (n = 2, 7%) (**Fig 2**). The site of possible contamination was exclusively the coastal region (100%) for *L. (V.) naiffi* and mostly the Maroni region for *L. (V.) lainsoni* (60%) (**Fig 2**).

HIV serology was positive for 2 patients infected with *L. (V.) lainsoni* but for none of the patients infected with *L. (V.) naiffi*. These two patients had several large lesions up to 4cm and 3cm respectively.

The median duration of symptoms before diagnosis was two months [range 0.5–59]. The lesions were mainly ulcers (80%), involving uncovered areas. *L. (V.) lainsoni* seems to be more frequently responsible for multiple lesions (11 cases) than *L. (V.) naiffi* (0 case). Indeed, for *L. (V.) naiffi*, a mild clinical course was usually observed, with no regional lymph node; only one lesion per patient was described, involving exclusively the limbs. Socio-demographic and clinical characteristics of patients infected by *L. (V.) naiffi* and *L. (V.) lainsoni* are detailed in **S1** and **S2 Tables** respectively. There was no case of mucosal involvement in the study population.

 

**Table 1. Main findings according to *Leishmania* species.**

| Characteristics | Total n = 30 | *Leishmania naiffi* n = 5 | *Leishmania lainsoni* n = 25 |
|---|---|---|---|
| **Sex, n (%)** | | | |
| Male | 18 (60) | 5 (100) | 13 (52) |
| Female | 12 (40) | 0 (0) | 12 (48) |
| **Place of Birth, n (%)** | | | |
| French mainland | 5 (17) | 3 (60) | 2 (8) |
| French Guiana | 5 (17) | 1 (20) | 4 (16) |
| French Caribbean | 1 (3) | 1 (20) | 0 (0) |
| Brazil | 17 (57) | 0 (0) | 17 (68) |
| Surinam | 0 (0) | 0 (0) | 0 (0) |
| Other | 1 (3) | 0 (0) | 1 (4) |
| Unknown | 1 (3) | 0 (0) | 1 (4) |
| **Occupation, n (%)** | | | |
| Gold miner | 6 (20) | 0 (0) | 6 (24) |
| Forester | 1 (3) | 0 (0) | 1 (4) |
| Military | 1 (3) | 1 (20) | 0 (0) |
| Other | 13 (43) | 4 (80) | 9 (36)* |
| Unknown | 9 (30) | 0 (0) | 9 (36) |
| **Age at diagnosis** | | | |
| Median (range) (y) | 33,5 (0–63) | 39 | 31 (0–63) |
| < 16, n (%) | 5 (16.7) | 0 (0) | 5 (20) |
| ≥16, n (%) | 25 (83.3) | 5 (100) | 20 (80) |
| **Site of probable contamination** | | | |
| Coastal region | 9 (30) | 5 (100) | 4 (16) |
| Maroni region | 15 (50) | 0 (0) | 15 (60) |
| Center region | 4 (13) | 0 (0) | 4 (16) |
| Oyapock region | 2 (7) | 0 (0) | 2 (8) |
| **Time of evolution before diagnosis, median (range) (months)** | 2 (0.5–59) | 2 (1–59) | 1.75 (0.5–7) |
| **Type of lesion, n (%)** | | | |
| Ulcer | 24 (80) | 4 (80) | 20 (80) |
| Papule | 0 (0) | 0 (0) | 0 (0) |
| Nodule | 3 (10) | 1 (20) | 2 (8) |
| Plaque | 0 (0) | 0 (0) | 0 (0) |
| Unknown | 3 (10) | 0 (0) | 3 (12) |
| **Number of lesions, n (%)** | | | |
| Unique | 17 (57) | 5 (100) | 12 (48) |
| Two | 7 (23) | 0 (0) | 7 (28) |
| ≥3 | 4 (13) | 0 (0) | 4 (16) |
| Unknown | 2 (7) | 0 (0) | 2 (8) |
| **Localization, n (%) ** | | | |
| Head/Neck | 4 (12) | 0 (0) | 4 (16) |
| Trunk | 0 (0) | 0 (0) | 0 (0) |
| Upper limbs | 14 (42) | 3 (60) | 12 (48) |
| Lower limbs | 14 (42) | 2 (40) | 11 (44) |
| Unknown | 1 (3) | 0 (0) | 1 (4) |
| **Regional lymph node, n (%)** | 7 (23) | 1 (20) | 6 (24) |
| **Lymphangitis, n (%)** | 2 (7) | 0 (0) | 2 (8) |
| **First line treatment, n (%)** | | | |

*(Continued)*

**Table 1.** (Continued)

| Characteristics | Total n = 30 | *Leishmania naiffi* n = 5 | *Leishmania lainsoni* n = 25 |
|---|---|---|---|
| Abstention | 5 (17) | 3 (60) | 2 (8) |
| Pentamidine | 16 (53) | 2 (40) | 14 (56) |
| Meglumine antimoniate | 1 (3) | 0 (0) | 1 (4) |
| Liposomal amphotericin | 0 (0) | 0 (0) | 0 (0) |
| Unknown | 8 (27) | 0 (0) | 8 (32) |
| **Spontaneous healing** | 4 (12) | 3 (60) | 1 (4) |

\* 5 patients were children; others included one teacher, one medical resident, one was build worker, and one patient had no job

\*\* Some patients had multiples lesions

## Therapeutics and follow-up data

**L. (V.) lainsoni.** The first line therapy was pentamidine for 14 (56%) of patients, meglumine antimoniate (mean dose of 20 mg antimony/kg/day) for one and abstention for two patients. During the follow-up, six patients (35%) required a second course of therapy (new dose of pentamidine for five patients and switch to meglumine antimoniate for one (**Table 1**; **S2 Table**).

**Leishmania (Viannia) naiffi.** Two subjects were initially treated with pentamidine and three individuals were initially not treated. One patient experienced a poor response to pentamidine therapy and required a second line by meglumine antimoniate (mean dose of 20 mg antimony/kg/day) therapy (**Table 1**; **S2 Table**).

## A high rate of pediatric cases was observed with *L. (V.) lainsoni*

*L. (V.) lainsoni* presented a high rate of pediatric cases. Indeed, five patients (20%) were children (<16 years old) (**Table 1**), with three children under two years old. They were

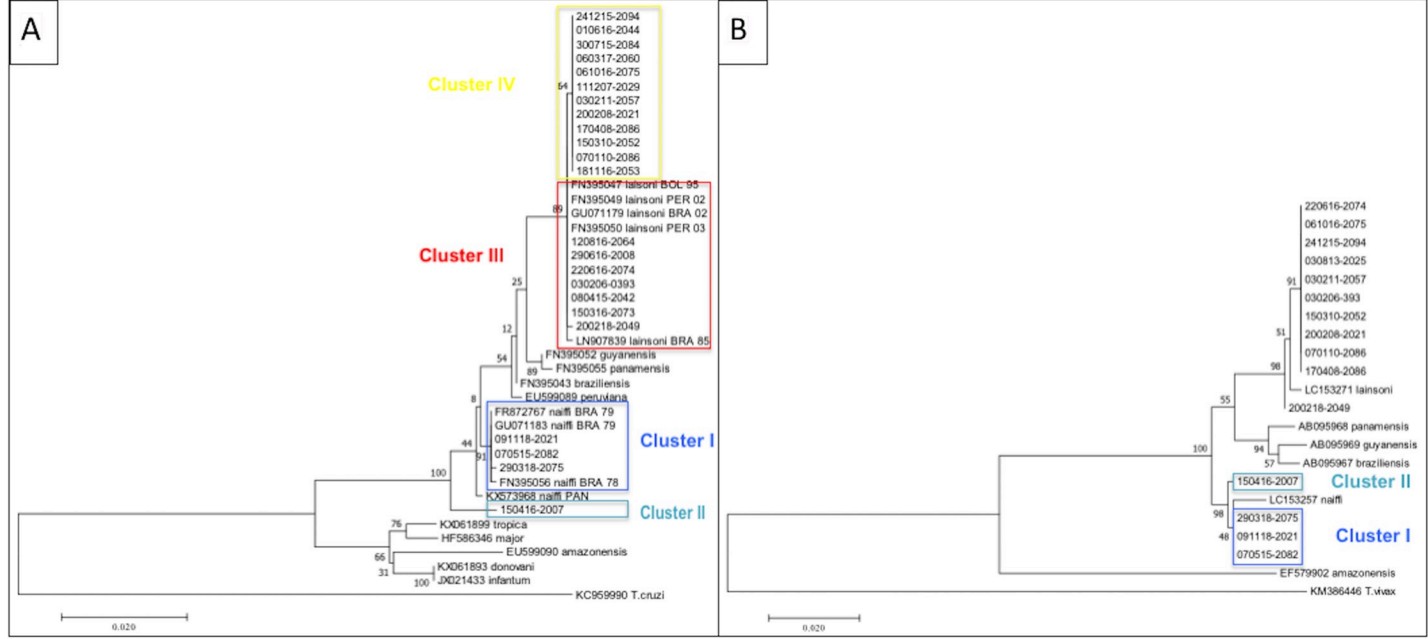

**Fig 2. Geographical distribution of the different genotypes within the 18 L. (V.) lainsoni and the 4 L. (V.) naiffi in French Guiana (drawn with http://glovis.usgs. gov/) Each dark blue circle represents one patient from the cluster I.** The light blue circle represents the only patient from cluster II. The exact contamination site of the only patient from cluster II is uncertain, but is located somewhere in the primary forest of the coastal region.

predominantly born in French Guiana (4/5) and one in Brazil. Three Boys and two girls were affected. They lived preferentially in rural areas, two on gold mining sites near the Maroni river, one at Saint-Elie, one at Cacao and the last one at Saint-Laurent. Their places of living were the presumptive contamination sites. Children presented a median of one lesion per patient [range: one to three]. Time-to-diagnosis was less than three months for all children. Ulcers were the unique lesion type with presence of regional lymph node in two patients. Two children presented lesions located on the head. Among the five children, three were initially treated with a unique pentamidine injection with a healing of cutaneous lesions. The youngest child was not treated (with self-disappearance of lesions) and the last one was lost to follow-up.

### *HSP70* and *cyt b* partial nucleotide sequence analysis

Though initial cultures could not be retrieved due to logistical issues, properly stored DNA samples were retrieved for all patients with *L. naiffi* and *L. lainsoni*. These samples were then used for PCR targeting HSP70 and *cytb*. Concerning HSP70, amplification was achievable for four samples of *L. (V.) naiffi* and 18 samples of *L. (V.) lainsoni* (**Fig 1**). Concerning *cyt b*, amplification was achievable for 4 patients with *L. (V.) naiffi* and 11 samples of *L. (V.) lainsoni* (**Fig 1**)

Sequences obtained for both genes were aligned and a robust phylogenetic reconstruction was carried out [35,36]. The results with *HSP70* and *cyt b* showed differentiated clusters corresponding to *L. (V.) naiffi* and *L. (V.) lainsoni*, which were also separated from all other *Leishmania* species (**Fig 3**). Intra-species divergence was also observed (**Fig 3**). The common term of haplotype is a specific group of mutations or a collection of Single Nucleotide Polymorphism (SNP) in the orthologous gene of the parasite. Diversity analysis for HSP70 was performed on 24 sequences of *L. lainsoni* and 8 sequences of *L. naiffi*, showing 3 and 23 polymorphic sites and 3 and 23 mutations respectively (**Table 2**). Based on the haplotype (Hd) and nucleotide diversity indexes (π), for each species, *L. (V.) naiffi* showed a higher genetic diversity (Hd = 0,893) and nucleotide diversity (π = 0,00519), than *L. (V.) lainsoni*, which showed a moderate genetic diversity (Hd = 0,598) and nucleotide diversity (π = 0,00055) (**Table 2**). Regarding the *cyt b* gene, haplotype and nucleotide diversity indexes also revealed that *L. (V.) naiffi* had higher values (Hd = 0,700, π = 0,00357), compared to *L. (V.) lainsoni* (Hd = 0,318, π = 0,00132) (**Table 2**).

For *L. (V.) naiffi*, the sequences obtained with *HSP70* were closed to Brazilian strains (cluster I). We noticed that patient n˚2 corresponding to cluster II (light blue shading) diverged from the others and from the Brazilian strains, individualizing two different genotypes within *L. (V.) naiffi* (**Fig 3A**). Analysis with *cyt b* gene supported these results (**Fig 3B**).

For *L. (V.) lainsoni*, phylogenetic analysis performed with *HSP70* gene seemed to highlight two profiles, profile III (red outlining) constituted by seven strains and profile IV (yellow outlining) constituted by twelve strains (**Fig 3**). These two groups were distinguished by a mutation concerning a single nucleotide c.675G>A. Analysis with *cyt b* did not yield any significant results due the several amplification failures (**Fig 3B**). As these two groups were distinguished by a single mutation of HSP70, they might not entirely comply with the definition of "cluster". However, it is worth noticing that profile III included many reference samples from Peru, Brazil and Bolivia. Conversely, strains from profile IV all originated from French Guiana. Patients from profile III were contaminated along the Maroni and the Oyapock rivers for five and two patients respectively. In group IV, the site of probable contamination was the coastal region and the center region for four and two patients respectively, and the Maroni river for five patients. The sites of probable contamination of all different clusters are detailed in **Fig 2**.

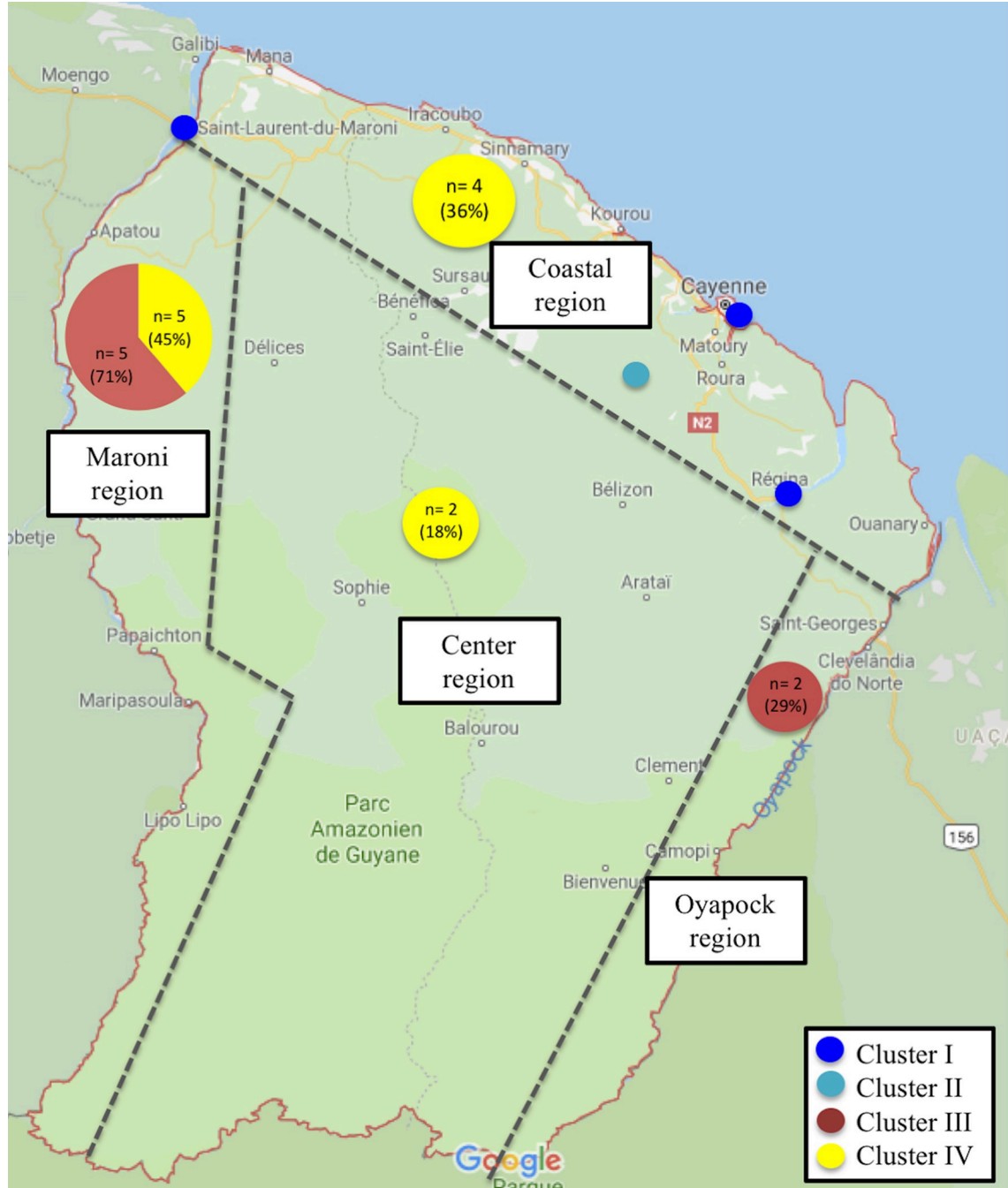

**Fig 3. Molecular *HSP70* (A) and *cyt b* (B) partial nucleotide sequence analysis by Maximum Likelihood method** The evolutionary history was inferred by using the Maximum Likelihood method based on the Tamura-Nei model [35]. The tree with the highest log likelihood (-3176.92) is shown. The percentage of trees in which the associated taxa clustered together is shown next to the branches. Initial tree(s) for the heuristic search were obtained automatically by applying Neighbor-Join and BioNJ algorithms to a matrix of pairwise distances estimated using the Maximum Composite Likelihood (MCL) approach, and then selecting the topology with superior log likelihood value. The tree is drawn to scale, with branch lengths measured in the number of substitutions per site. The analysis involved 42 nucleotide sequences. Codon positions included were 1st+2nd+3rd+Noncoding. All positions containing gaps and missing data were eliminated. There were a total of 1242 positions in the final dataset. Evolutionary analyses were conducted in MEGA7 [36]. Dark blue shading represents patients from cluster I; Light blue shading represents patients from cluster II. Red shading indicates patients from cluster III and yellow shading indicates patients from cluster IV.

**Table 2. Genetic diversity parameters of *Leishmania HSP70* and *Cyt b* genes sequences.**

| HSP70 gene | | | | | | |
|---|---|---|---|---|---|---|
| Species | N | S | Eta | Hd | π | K |
| *L. (V.) Lainsoni* | 24 | 3 | 3 | 0,598 | 0,00055 | 0,688 |
| *L. (V.) naiffi* | 8 | 23 | 23 | 0,893 | 0,00519 | 6,464 |
| Cyt b gene | | | | | | |
| Species | N | S | Eta | Hd | π | K |
| *L. (V.) Lainsoni* | 12 | 5 | 5 | 0,318 | 0,00132 | 1,106 |
| *L. (V.) naiffi* | 5 | 7 | 7 | 0,700 | 0,00357 | 3,000 |

N = Number of sequences. S = Number of polymorphic sites.

Eta = Total number of mutations.

Hd = Haplotype diversity. π = Nucleotide diversity.

K = Average number of nucleotide differences.

## Discussion

So far, *Leishmania lainsoni* and *naiffi* are the least studied species of the *Leishmania* genus in the New World. This study offers new data concerning the clinical and epidemiological features of patients infected with these species in French Guiana.

French Guiana is divided in two different ecological regions, the hinterland covered by primary rain forest, where small gold mines and Amerindian villages are the only human settlements. On the other hand, the coastal region covers the north of the territory and is more anthropized [6,37,38]. Interestingly, our results suggest distinct socio-epidemiological features for *L. (V.) naiffi* and *L. (V.) lainsoni*. The analysis of species at the regional level showed different patterns of geographic distribution between *L. (V.) naiffi* and *L. (V.) lainsoni*. In fact, patients seemed to get infected with *L. (V.) naiffi* during leisure activities in anthropized coastal areas, while *L. (V.) lainsoni* seemed to be acquired during professional activities (such as gold mining) in primary forest regions [37]. *L. (V.) lainsoni* was also reported mainly in rain forest regions in Peru [39].

Besides, our cohort of *L. (V.) naiffi* comprised no Brazilian patient and no gold miner, though Brazilian gold miners are known to make up most of the population of CL in French Guiana [5,38]. On the other hand, infections with *L. (V.) lainsoni* involved Brazilian patients in 68% of cases. Only 24% introduced themselves as gold miners, but this activity was probably under-reported, as it is illegally performed on French territory. Therefore, the epidemiology of *L. (V.) lainsoni* in French Guiana seems very similar to more frequent species like *L. (V.) guyanensis* or *L. (V.) braziliensis* [4,5,7,40].

A mild clinical course was observed for all cases of *L. (V.) naiffi* in this study. These findings are supported by experimental studies showing that *L. (V.) naiffi* frequently causes mild or even non-apparent infections on hamsters' skin [9]. In vitro analysis also demonstrated that *L. (V.) naiffi* had the lowest infection index and the highest nitric oxide production compared with other species of the *Viannia* subgenus and consequently was possibly less pathogenic in human [41]. *L. (V.) naiffi* infections could also be under-diagnosed, as patients with self-limited skin lesions may not seek consultation. Besides, the introduction of new identification techniques such as gene sequencing improves the precision of species isolation and increases the likeliness of identifying new cases of *L. naiffi* infections.

Interestingly, *L. (V.) Lainsoni* appears to be associated with a high rate of pediatric cases. In the literature, only one case of pediatric CL caused by *L. (V.) Lainsoni* was reported in Manaus, Brazil [42]. Information on disease burden, clinical spectrum and efficacy of treatment in

pediatric CL remains scarce. In our cohort, 3/5 children lived in particular exposed areas, near gold mining sites with active deforestation drawing vectors closer to human dwellings [43]. The presence of an intra-familial CL case was found to be an important risk factor in the literature [44]. However, there was no element in our data to support the hypothesis of intra-domiciliary contamination, as all patients belonged to different families. However, ancient infections in parents could have been unnoticed. Concerning the other clinical features of *L. (V.) lainsoni*, our results were similar to previous reports, the predominant presentation being a single ulcer, localized on the face or the extremities [43,45,46]. Among the 4 children with available data on treatment, one had a spontaneous healing and three were cured with pentamidine, contrasting with several studies reporting high proportions of treatment failure [47–49]. Our data suggest that pentamidine could be an adequate treatment for pediatric CL with *L. (V.) lainsoni.*

Concerning therapeutic data in adults, only one patient with *L. (V.) naiffi* presented a therapeutic failure with pentamidine and required meglumine antimoniate. This patient had specific clinical features with a chronic leishmaniasis presentation. The other patients had benign clinical course with spontaneous healing or good response to pentamidine, as reported in several series [12–15]. Recent reports suggested that *L. (V.) naiffi* could be resistant to firs-line treatment, with a poor response to meglumine antimoniate or pentamidine [16,17]. Goncalves et al. [17] discussed the possible association between poor response and the presence of Leishmania RNA virus that could contribute to increased parasite virulence by reducing the sensitivity to oxidative stress [50]. Further studies are needed to understand the factors associated with treatment failure.

Regarding *L. (V.) lainsoni*, therapeutic data in the literature are lacking. In our cohort, the first line therapy was pentamidine for 14 (56%) of patients. Among these patients, 10 were cured or presented a partial response, one had no improvement and data were missing for three patients. Our data seems to indicate a pentamidine-sensitive disease.

*HSP70* and *cyt b* partial nucleotide sequence analysis of 4 *L. (V.) naiffi* infections and 18 cases of L. *lainsoni* offers new insight on their intra-species genetic variability. Sequencing of these two targets identified higher intra-species variability in *L. (V.) naiffi* (**Table 2**). In a previous study, estimates of divergence among five different species also identified *L. (V.) naiffi* as the most polymorphic one [51]. However, these findings should be confirmed with a higher number of samples.

When examining possible associations between genotypic groups and epidemiological or geographical data, several links could be observed. The only patient from cluster II (patient 2) belonged to the military and consequently worked in primary forest and might have been infected on his workplace, whereas patients from cluster I were contaminated in the coastal region during leisure activities. These two distinct *L. (V.) naiffi* populations could reflect the role played by two different ecological regions with different vectors and hosts. Indeed, the costal region is typically divided in multiple ecotopes such as beaches, mangroves, coastal swamps and small patches of primary rainforests. It has been reported in Ecuador that peridomiciliary ecotopes were characterized by higher diversity and number of infected sand flies than forest ecotopes [52] due to the high diversity of blood sources available on the coast (such as humans, chickens, cows, and dogs) [53]. It is therefore essential to determine if domesticated animals are reservoirs of CL in French Guiana, as the risk of infection would be higher in these coastal areas than in the forest of the hinterland.

While patient 2 remained separated, samples from Cluster I were closely related to previously published Brazilian strains. However, clinical features of our patients could not be compared with previous works as these studies did not mention any symptom or therapeutic response [51,54,55].

Concerning *L. (V.) lainsoni*, the definition of genotypic groups must be taken with caution, as groups III and IV were distinguished by a mutation of a single nucleotide. Besides, results from HSP 70 sequencing were not supported by analysis of *cyt b*. HSP70 analysis was solely used for group definitions due to the less efficient DNA amplification with *cyt b* and the several amplification failures. Group III strains were similar to their counterparts from Peru, Brazil and Bolivia, while group IV was made up only by French Guiana strains. We noticed that in group IV, the reported site of contamination was the coastal region and the center region in four and two patients respectively, whereas these regions were never involved in patients from group III. The latter were all infected along the Maroni and the Oyapock, these two rivers being the natural borders between French Guiana, Surinam and Brazil. Therefore, a genetic proximity with Brazilian strains could be expected. Indeed, it has been suggested that a degree of similarity was higher between strains whose geographical origins were closer [56]. Unfortunately, there was no available sequence from Surinam to allow a genetic comparison of the Maroni strains. These results may suggest the existence of specific strains of French Guiana (group IV), along with a trans-border circulation of less specific ones (group III). However, we report no clinical, epidemiological or therapeutic difference between these two clusters of *L. (V.) lainsoni.*

However, the fact remains that the different clinical expressions of CL depend on both intra-species genetic diversity of *Leishmania* and host immune status [57]. In addition to the intra-species genetic variability of *Leishmania*, host immune reaction performs a significant function in the clinical presentation of CL.

Among *Viannia* parasites, reports have indicated that genetic variations were extensive, with some clones widely distributed and others localized to a particular endemic focus, with specific transmission cycles, probably reflecting an adaptation of different clones to the vector species involved [58,59]. Other results have suggested that their distribution was related to the origin of the gene pool as well as to present vector and reservoir movements [60]. The genetic polymorphism among strains of *L. (V.) naiffi* and *L. (V.) lainsoni* could be associated with the eco-epidemiology of the areas, as reported for *L. (V.) braziliensis* [59,60].

Epidemiological features reported in his study suggest that *L. (V.) naiffi* and *L. (V.) lainsoni* circulate in two ecologically different regions. Indeed, *L. (V.) naiffi* has been associated with *Lu. squamiventris* in French Guiana [61], collected in the savanna area of Sinnamary [61], in the coastal region, which reinforce our results and our idea that *L. (V.) naiffi* transmission occurs mostly on the coast of French Guiana. *L. (V.) lainsoni* was first isolated from a wild mammal in 1991, from the Agouti paca (*Rodentia*: *Dasyproctidae*), in the state of Pará, Brazil [62]. Natural infections with *L. (V.) lainsoni* in sand flies have been detected in *Lu. ubiquitalis* in primary forest of Brazil and Ecuador [53,63,64], in *Lu. nuneztovari anglesi* in Bolivia [20], and in *Lu. Auraensis*, particularly abundant in the southeastern rainforest in Peru [65]. Though vectors of *L. (V) lainsoni* have not yet been identified in French Guiana, reports from neighboring countries offer explanations to our epidemiological features, *L. (V.) lainsoni* being well adapted to primary forest environments and therefore likely to infect miners working in gold camps in the rainforest.

There are several limitations to our study. We conducted a retrospective inclusion and the number of our patients remained small, which did not allow us to perform a thorough statistical analysis. Besides, clinical data of patients seen in remote health centers were provided by general practitioners and not by dermatologists, inducing a possible bias. Missing data regarding outcome after treatment limited our results. Indeed, most of illegal gold miners are Brazilian, from the poorest states of Brazil with difficult access to health care [66]. Finally, correlations between clinical data and genetic features must always be interpreted with caution, particularly when dealing with small-size samples.

Despite these limitations, this study reports important clinico-epidemiological features of these two rare *Leishmania* species and bring new insight into the intra-species genetic variability. To our knowledge this is the largest study covering the cases of *L. (V.) lainsoni* and the first study analyzing the genetic diversity of these species in French Guiana. These findings will be of great help to enhance the management of CL to *L. (V.) naiffi* and *L. (V.) lainsoni* in French Guiana and to develop new studies. For further studies, it is mandatory to conduct Whole Genome or Exome Sequencing of these parasites to obtain a suitable picture of intra-species variability.

## Conclusion

*L. (V.) naiffi* and *L. (V.)* lainsoni are significantly present in French Guiana and probably across South America. Our data suggest distinct socio-epidemiological features for these two *Leishmania* species. Patients seem to get infected with *L. (V.) naiffi* during leisure activities in anthropized coastal areas, while *L. (V.) lainsoni* shares common clinical and epidemiological features with *L. (V.) guyanensis and braziliensis* and seem to be acquired during professional activities in primary forest regions. This study provides an example of associations between epidemiological features and different *Leishmania* species. The improvement of laboratory techniques for species identification in South America must be used to constantly update and improve our knowledge of different parasitic species, as these data may help to target specific populations in public health actions. Determination of vector–parasite–reservoir relationships is needed to understand the transmission dynamics of these species and implement control strategies. The inclusion of more patients across South America and the generalization of identification techniques such as gene sequencing or mass spectrometry could allow the constitution of larger cohorts and allow us to confirm suggested clinical features of *L. naiffi* and *L. lainsoni*.

## Supporting information

**S1 Checklist. STROBE Checklist.**
(DOCX)

**S1 Table. Demographic and clinical data of patients infected with *Leishmania (Viannia) naiffi*.**
(DOCX)

**S2 Table. Demographic and clinical data of patients infected with *Leishmania (Viannia) lainsoni*.**
(DOCX)

## Author Contributions

**Conceptualization:** Romain Blaizot.

**Data curation:** Ghislaine Prévot.

**Investigation:** Océane Ducharme, Stéphane Simon, Marine Ginouves, Ghislaine Prévot, Romain Blaizot.

**Methodology:** Magalie Demar, Romain Blaizot.

**Resources:** Ghislaine Prévot.

**Supervision:** Pierre Couppie, Magalie Demar.

**Validation:** Romain Blaizot.

**Writing – original draft:** Océane Ducharme.

**Writing – review & editing:** Magalie Demar, Romain Blaizot.

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
