## [Decision Letter · Decision Letter 0]

22 Feb 2020

Dear Dr Blaizot,

Thank you very much for submitting your manuscript "Leishmania naiffi and lainsoni in French Guiana: clinical features and phylogenetic variability" for consideration at PLOS Neglected Tropical Diseases. As with all papers reviewed by the journal, your manuscript was reviewed by members of the editorial board and by several independent reviewers. In light of the reviews (below this email), we would like to invite the resubmission of a significantly-revised version that takes into account the reviewers' comments. 

We cannot make any decision about publication until we have seen the revised manuscript and your response to the reviewers' comments. Your revised manuscript is also likely to be sent to reviewers for further evaluation.

Sincerely,

Ikram Guizani

Guest Editor

Jesus Valenzuela

Deputy Editor

Reviewer's Responses to Questions

**Key Review Criteria Required for Acceptance?**

**Methods**

-Are the objectives of the study clearly articulated with a clear testable hypothesis stated?

-Is the study design appropriate to address the stated objectives?

-Is the population clearly described and appropriate for the hypothesis being tested?

-Is the sample size sufficient to ensure adequate power to address the hypothesis being tested?

-Were correct statistical analysis used to support conclusions?

-Are there concerns about ethical or regulatory requirements being met?

Reviewer #1: This is a descriptive study showing several characteristics associated with cutaneous leishmaniasis caused by L. lainsoni and L. naiffi in French Guiana. Although the study is focusing in French Guiana, the similarity observed for strains of each species with strains from other localities indicates that the discussion presented is relevant for the understading of the disease caused by both parasites.

Reviewer #2: The methods are clearly described and adequate. The sample size is low but this is inherent to the study of a very rare disease; and actually, the cohort is of an interesting size for the Leishmania species studied.

It is written that "diagnosis" was performed using PCR-RFLP, hsp70 sequencing or MALDI-TOF. I understand that species typing was done using these methods, but not diagnosis. Could the authors specify how the initial/primary diagnosis was made (first arrow of Fig.1)?

The in vitro cultivation method is not indicated.

This reviewer is not competent to assess the validity of the phylogenetic methods and inferences.

**Results**

-Does the analysis presented match the analysis plan?

-Are the results clearly and completely presented?

-Are the figures (Tables, Images) of sufficient quality for clarity?

Reviewer #1: It is important to clarify the identification of L. naiffi parasites. It was mentioned that all L. naiffi cases resulted in positive cultures, but one failed for hsp70 PCR. Why? I can't see a reason for PCR hsp70 PCR fail when DNA from Leishmania culture is employed. 

It is better to try to re-write the topic identification of Leishmania species as this is a bit confused to follow.

The fact that L. lainsoni was frequently more associated with patients born in Brazil is correctly associated with professional activity as gold mining. It is already known the association between Brazilian gold miner and CL in French Guiana. I suggest not to link L. lainsoni patients to the fact that they were born in Brazil, as what is significant here is the professional activity.

Why for L. lainsoni all the socio-demographic and clinical characteristics were presented in general in Table 1 and for L. naiffi a detailed table was presented for each patient? Table 2 should include information on L. lainsoni patients and presented as Supplementary Table.

Phylogenetic analysis 

Although it is large employed the term “phylogenetic analysis” for the analysis conducted in this study, my suggestion is to replace that for “HSP70 and cytb partial nucleotide sequence analysis”. The idea here was not to infer about evolutionary relationship, but to perform a good explanatory analysis employing distance based method. 

I suggest to present the analysis performed with HSP70 and cytb genes in the same figure, as A and B. The results from cytb were not completely concordant to hsp70 and this must be considered to define the groups, although the sequences analyzed are not exactly the same. For example, cluster IV considered as a group for L. lainsoni using hp70 sequences is not supported by bootstrap values in the hsp70 tree or by cytb analysis.

Reviewer #2: The Results are clearly and completely presented.

Although I understand that infections due to L. naiffi are particularly rare, the low number of cases due to L. naiffi is a problem for inferring any conclusions about differences between the two species. The sentence "L. (V.) lainsoni was more frequently responsible for multiple lesions (11 cases) than L. (V.) naiffi (0 case)" should be attenuated (e.g. "seems to be"). Statistical considerations made from 5 cases cannot be put forward. In that sense, the authors took care to not write that there were more paediatric infections due to L. naiffi than to L. lainsoni, but more cautiously, that 20% of infections due to L. lainsoni were observed in children. The same caution should be applied throughout the paper. 

The probable month of contamination should be indicated whenever possible for all cases. This is easy to calculate from the date of diagnosis and the delay between onset of symptoms and diagnosis. It would allow some considerations about the period of contamination for both species.

Spontaneous healing of the lesions is mentioned several times throughout the paper. It would be good if this was summarized for all cases with the clinical course addressed in page 11.

This reviewer is not competent to assess the validity of the phylogenetic methods and inferences. However, given the very low genetic distance observed between both clusters (evidenced by the existence of a single base mutation), the isolation of cluster IV versus cluster III may be a bit artificial? This distinction should perhaps be justified in the text.

**Conclusions**

-Are the conclusions supported by the data presented?

-Are the limitations of analysis clearly described?

-Do the authors discuss how these data can be helpful to advance our understanding of the topic under study?

-Is public health relevance addressed?

Reviewer #1: The authors should try to re-write this part. Some statements seem not adequate to be presented here. For example, this study does not allow concluding that “L. lainsoni seems to infect children more frequently than other species”. There is also some results presented as conclusion, like “A mild clinical course was observed for all cases of L. (V.) naiffi”. The second paragraph must be completely revised.

Reviewer #2: The conclusions of the study are well supported by the data presented. The limitations of the sudy are also well described. The Discussion provide an extensive literature search.

However, the Discussion is lengthy and would benefit from being shortened. For example, the considerations about the immune response in children (page 21) may be deleted. Also the Discussion rewrites several points of the Results, which appears redundant.

The Conclusion is redundant and may be deleted (except the sentence about "determination of vector–parasite–reservoir relationships "). The last sentence of the conclusion is useless (We showed the identification of different genotypes within L. (V.) naiffi and L. (V.) lainsoni and how these genotypes are distributed at the geographic level.).

**Editorial and Data Presentation Modifications?**

Reviewer #1: English / typing revision is required.

Reviewer #2: Abstract : "percentage is missing in "A high number of pediatric cases (n=5; %) was observed ".

Page 10: the sentence "Patients infected by L. (V.) lainsoni were significantly more frequently born in Brazil" is perturbing as it is placed in the middle of a paragraph exposing the whole of the data without considering the species.

Page 10-11: the regions mentioned in the text should refer to Figure 3 for the reader who does not have a good knowledge of French Guiana: it would help to a better understanding of the geographical origin of the cases mentioned here.

Discussion : first paragraph: remove "comprehensive", just write "new data".

A few minor English errors in the Abstract , Introduction, Fig. 1, Results etc. (have been/were, was/has been, consultation, theses, register, a second therapeutic line by a new injection…).

Note: the absence of line numbering makes it difficult (or time consuming) to address very minor points such as English errors.

**Summary and General Comments**

Reviewer #1: All comments were presented in the previous topics.

Reviewer #2: This manuscript addresses the question of cutaneous leishmaniosis due to two rare species of Leishmania in French Guiana. Although, precisely due to the scarcity of such infections, the number of cases included in the study is limited, the authors provide new data about clinical and epidemiological features of CL due to L. (V.) naiffi and L. (V.) lainsoni in French Guiana. They also show novel phylogenetic data about the genetic diversity between and within both Leishmania species. As such, the manuscript is interesting and worth publishing.

Yet, the authors should address a number of comments before publication (see other sections above).

PLOS authors have the option to publish the peer review history of their article (what does this mean?). If published, this will include your full peer review and any attached files.

Reviewer #1: No

Reviewer #2: No
---

## [Decision Letter · Decision Letter 1]

11 May 2020

Dear Dr Blaizot,

We are pleased to inform you that your manuscript 'Leishmania naiffi and lainsoni in French Guiana: clinical features and phylogenetic variability' has been provisionally accepted for publication in PLOS Neglected Tropical Diseases.

Best regards,

Ikram Guizani

Guest Editor

Jesus Valenzuela

Deputy Editor

The authors have well complied with the recommendations of the reviewers.

One of them however pointed to the fact that the data are not fully available as there is no accession numbers to all the sequences that were generated through this study.

Therefore, the final manuscript should include the accession numbers of all the sequences generated through this study.

Reviewer's Responses to Questions

**Key Review Criteria Required for Acceptance?**

**Methods**

-Are the objectives of the study clearly articulated with a clear testable hypothesis stated?

-Is the study design appropriate to address the stated objectives?

-Is the population clearly described and appropriate for the hypothesis being tested?

-Is the sample size sufficient to ensure adequate power to address the hypothesis being tested?

-Were correct statistical analysis used to support conclusions?

-Are there concerns about ethical or regulatory requirements being met?

Reviewer #1: NA

Reviewer #2: Fine

**Results**

-Does the analysis presented match the analysis plan?

-Are the results clearly and completely presented?

-Are the figures (Tables, Images) of sufficient quality for clarity?

Reviewer #1: NA

Reviewer #2: Fine

**Conclusions**

-Are the conclusions supported by the data presented?

-Are the limitations of analysis clearly described?

-Do the authors discuss how these data can be helpful to advance our understanding of the topic under study?

-Is public health relevance addressed?

Reviewer #1: NA

Reviewer #2: The Discussion is still a bit lengthy but has been improved; and the authors have discussed their data in a relevant manner. I would simply delete the word "CONCLUSION" (if not mandatory) and delete the 4 first sentences of this Conclusion to merge this paragraph with the previous one, which already looks like a conclusion.

**Editorial and Data Presentation Modifications?**

Reviewer #1: NA

Reviewer #2: Just a few typos: Table 1, line 375, line 376: Lainsoni should not be written with a capital; line 302: "closed" should read "close"; line 310: delete "the"; line 455: "his"; .…

**Summary and General Comments**

Reviewer #1: The authors modified the manuscript accordingly to the suggestions made by both reviewers and now I think it is suitable for publication at PNTD.

Reviewer #2: The authors have responded correctly to my comments, and the MS is improved.

PLOS authors have the option to publish the peer review history of their article (what does this mean?). If published, this will include your full peer review and any attached files.

Reviewer #1: No

Reviewer #2: No

---

## [Editor Report · Acceptance letter]

21 Jul 2020

Dear Dr Blaizot,

We are delighted to inform you that your manuscript, "*Leishmania naiffi* and *lainsoni* in French Guiana: clinical features and phylogenetic variability," has been formally accepted for publication in PLOS Neglected Tropical Diseases.

Best regards,

Shaden Kamhawi

co-Editor-in-Chief

Paul Brindley

co-Editor-in-Chief
